# Two-Stage 3D Codebook Design and Fast Beam Search Algorithm for Millimeter-Wave Massive MIMO Systems

**Zhangyou Peng** *,† and **Wen Li** † 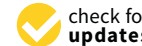

Key Laboratory of Specialty Fiber Optics and Optical Access Networks, Shanghai University,
Shanghai 200444, China; vector@t.shu.edu.cn
* Correspondence: zypeng@i.shu.edu.cn or zypeng@mail.shu.edu.cn
† These authors contributed equally to this work.

**Abstract:** For 3D beam access in massive multi-input multi-output (MIMO) systems, this paper proposes a feasible two-stage 3D codebook design and corresponding fast beam training algorithm. The complete codebook consists of a primary codebook and an auxiliary codebook—the primary codebook produces a basic directional beam using a smaller number of phase shifts to maintain lower hardware complexity. The small auxiliary codebook creates finer beams centered on each main beam. In addition, a beam search scheme for two-level codebook design is proposed. The main beam search is performed using a modified binary search algorithm based on the signal-to-noise ratio threshold; with the refined auxiliary codebook, the auxiliary beam search is completed using a basic binary search. The simulation results verify the performance advantages of our proposed codebook in terms of beamforming gain and beam access accuracy in MIMO systems. In addition, the comparison of beam training complexity shows that our specially designed two-stage beam search scheme has significant advantages over other existing solutions.

**Keywords:** millimeter-wave; analog beamforming; two-stage codebook; 3D fast beam search

## 1. Introduction

Recent years, in the field of mobile wireless communications, millimeter-wave (mm-Wave) communication has received extensive attention due to its extremely wide bandwidth spectrum resources and high transmission rate [1]. In millimeter-wave communication, digital baseband beamforming is extremely difficult in massive multiple-input multiple-output (MIMO) systems because of high power consumption [1]. Instead, analog beamforming (ABF) can further reduce transmission attenuation because of its simplicity and utility [2]. ABF can generally be divided into two categories: non-codebook adaptive beamforming [3] and codebook-based switched beamforming [4,5]. Adaptive beamforming uses an adaptive algorithm to adjust the weight of each array element, but it requires complete channel state information (CSI), which is difficult to implement. Instead, the switching beamforming strategy detects on a predetermined codebook, and the best beam pair can be found without CSI. Therefore, the switching beamforming method is more practical and the most important problem in switched beamforming is the design of codebook and beam training algorithms to reduce beam search time and improve received signal-to-noise ratio (SNR). There has been a lot of related work in this area. For codebook design, the IEEE 802.15.3c codebook [4] uses only four phase shifts per element to reduce complexity. The DFT codebook in Reference [5] has more phase shifts and achieves higher array gain. In addition, a beam control codebook is proposed in Reference [6], which enlarges the size of the codebook by performing finer quantization on the beam angle. In order to

improve search efficiency, we can define hierarchical codebooks that generate beams of different resolutions [7–9]. For the 3D codebook design, Reference [9] envisions a simple binary tree codebook structure and corresponding fast beam search algorithm. The literature [10] proposes a KPC codebook which generates the final codebook by using the Cronecker product of the horizontal codeword and the vertical codeword. This method improved spatial degrees of freedom to match 3D MIMO channel characteristics and to improve system performance. Ref. [11] proposed a beam pattern synthesis method to synthesize a rectangular beam pattern. Another solution based on two-dimensional antenna arrays is to use Hierarchical codebook [12], and the author of Reference [12] proposed a layered 3D codebook design, which can be well adapted to the 3D beam training algorithm proposed in the paper, and the traditional 3c and DFT codebooks are extended to 3D for comparison.

In addition, many corresponding beam search algorithms have been proposed. Among them, the most direct algorithm is the exhaustive search, but it has extremely long beam search time [13,14]. Some existing literature also proposes some hierarchical or multi-level beam training schemes to shorten training time. At present, the IEEE 802.15.3c [4] standard proposes a two-level search scheme, that is, a sector level and an accurate beam level, and the search complexity can be proportional to the square of the antenna number (search complexity refers to the number of times the beam search is performed during signal access process, and the time of each probe is approximately constant). The search complexity in 802.11ad [15] can be reduced to a linear relationship with the number of antennas. Reference [9] considers a binary search like (BSL) scheme that reduces the search complexity to a logarithmic relationship with the number of antennas. The authors of Reference [16] proposed joint judgment based on the main lobe overlap state of all training beams, and mentioned some new concepts, such as state utilization efficiency (SUE) and feedback efficiency (FE), and explained the search complexity from the perspective of information theory. The author of the literature [12] adopted the idea of hierarchical search, extending the beam training to two dimensions of horizontal angle and elevation angle, and correspondingly achieved lower training complexity. Reference [12] applied a two-stage search method of coarse search and detailed search to reduce the beam search time and improve the signal-to-noise ratio, but the search process at each stage can be further simplified. Furthermore, the context-based information approach in References [17,18] utilizes global positioning systems (GPS) to resolve BS coordinates, but this kind of solution is limited to outdoor settings with permanent GPS connectivity. Nowadays some papers leverage analog beamforming techniques and proposes a new solution, termed as simultaneous multi-beam analog beamforming (MB-ABF). Reference [19] firstly presents a novel fast access scheme based upon simultaneous multi-beam beamforming and achieves one of the fastest initial access times. However, when achieving the same refined beamwidth, the total number of antennas in mobile devices is extremely very large and the antenna array spacing is larger, which limits the application of the algorithm. The necessary orthogonal data encoding also introduces additional energy consumption. The encoding and interference problems that multi-beam training brings to beamforming systems make such methods unsuitable 3D beam training scenarios. To briefly summarize the beam training algorithms mentioned above, referring to Table 1, we list the commonly used schemes and analyze the beam training complexity, accuracy and characteristics of different methods.

**Table 1.** Comparison of mmWave beam training schemes.

| Scheme | Complexity | Characteristic | Ref. |
|--------|-----------|---------------|------|
| Exhaustive search | prohibitively high | extremely high search time<br>extremely high beam search accuracy. | [13,14] |
| IEEE 802.11.15.3c | high | high search time<br>high search accuracy. | [4] |
| IEEE 802.11.ad | moderate | relatively high search time<br>relatively high search accuracy. | [15] |
| Hierarchical search | low | low training time<br>guaranteed search accuracy. | [9,12] |
| Context-Information based | low | extremely low search time<br>high search accuracy<br>needs permanent GPS connection. | [17,18] |
| Multi-beam search | low | extremely low search time<br>not suitable for 3D scenarios<br>moderate search accuracy. | [19] |

Most of the existing researches focus on the ULA (uniform linear array) model. Many classical traditional algorithms and codebooks can be extended to 3D beamforming, but lack of good performance and applicability. Many analog beamforming vectors are designd with fixed shape or beamwidth based on the ULA model. Due to the lack of flexibility and applicability, it is difficult to apply these designs to non-ULA [11,16]. In fact, most of the antennas deployed in cellular systems are URA (uniform rectangular array), which requires a more flexible framework that takes into account different antenna configurations. For 3D MIMO beam access scenarios, we need a new codebook design and beam access method to significantly reduce access latency and ensure reliable communication. This paper proposes a feasible two-stage 3D codebook design and corresponding fast beam training algorithm. By using the primary codebook, the primary search of the beam training algorithm can obtain a higher cumulative beam gain with lower time complexity; further using the refined secondary codebook, the auxiliary search can further improve the beam gain. The codebook design and algorithm ideas in this article are also applicable to the ULA model. In general, the main contributions of this paper are as follows:

- We considered different antenna array shapes, including ULA (this is the most common array shape in the literature) and URA (which is more practical and under-researched in the literature).
- We propose a general codebook design method to generate 3D beam codebooks with different resolutions, and apply to the corresponding beam access method.
- We propose a new 3D beam access method, and with the codebook design we verify the robustness of codebooks and algorithms using different metrics such as search complexity, accuracy and cumulative beam gain.

The remainder of this paper is organized as follows: In Section 2, we introduce the millimeter-wave channel model and system model used in this paper. In Section 3, the 3D two-stage codebook design will be described in detail. Section 4 implements the corresponding fast search algorithm. Section 5 presents the simulation verification of search complexity and accuracy. Finally, we made some summary of our work and made a prospect for future research directions in Section 6.

## 2. General Settings

### 2.1. Spatial Response

First, the 3GPP antenna model is described [for example, refer to TR38.901]. The antenna is defined in 3D coordinate system, as shown in Figure 1. The incoming ray is represented by a pair of

angles $(\theta, \phi)$, wherein the angle $\theta$ is elevation angle respect to the axis $z$, $\phi$ is the angle between the projection of the incident electromagnetic wave with respect to the plane $x - y$ and the axis $x$, that is, the azimuth angle. The URA is located in plane $x - y$ with a row of $N$ elements in the $x$ axial direction and $M$ elements in the $y$ axial direction. The entire array antenna array has a total of $N \times M$ array elements. The spacing between the antenna elements in the direction of the $x$ axis and the $y$ axis are $d_x$ and $d_y$ respectively. For a one-dimensional array, $M$ will be reduced to 1 and the array response will be the same in the direction $\theta$.

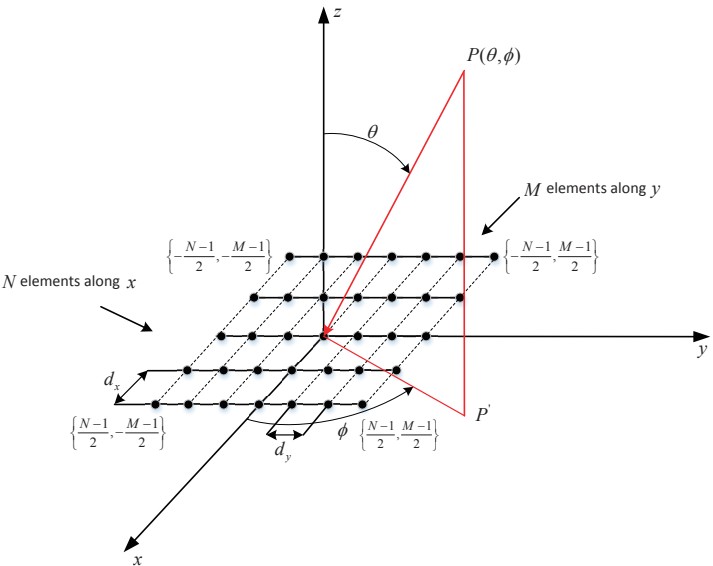

**Figure 1.** Uniform rectangular planar array model.

We define the antenna index $(m_x, m_y)$ as the $m_x - th$ antenna in $x$ axial direction and the $m_y - th$ antenna in $y$ axial direction. Considering the plane wave model and the isotropic element, the position of the antenna element introduces a phase offset. The antenna array element labeled as $(m_x, m_y)$ was applied a phase shift $w_{m_x m_y}$, the array response factor (AF) of the antenna system can be written as [12]:

$$AF(\theta, \phi) = \mathbf{w}^H \mathbf{a}(\phi, \theta)$$
$$= \sum_{m_x=0}^{N-1} \sum_{m_y=0}^{M-1} w_{m_x m_y} e^{j2\pi\left(\frac{m_x d_x}{\lambda} \sin\phi \sin\theta + \frac{m_y d_y}{\lambda} \cos\theta\right)}. \tag{1}$$

Among them, $\lambda$ is the wavelength, $\phi$ and $\theta$ are the azimuth and the elevation angle respectively. Typically, the antenna element spacing $d_x$ and $d_y$ are set to $\lambda/2$. The antenna array response vector is simplified and written as:

$$\mathbf{a}(\phi, \theta) = \left[1, ..., e^{j\pi\left(m_x \sin\phi \sin\theta + m_y \cos\theta\right)}, ..., e^{j\pi((N-1)\sin\phi \sin\theta + (M-1)\cos\theta)}\right]. \tag{2}$$

AF is expressed in a vector form: $AF(\phi, \theta) = \mathbf{w}^T(\phi, \theta) \mathbf{a}(\phi, \theta)$, and the weight vector is expressed as: $\mathbf{w}^T(\phi, \theta) = [w_0, w_1, ..., w_{NM-1}]$. Here, we need to design the AWV (array weighted vector) to obtain large beam gain as possible as can. For the two-stage 3D codebook design proposed in this paper, the weight vector is defined as:

$$\mathbf{w}(\phi, \theta) = \mathbf{w}(\phi) \otimes \mathbf{w}(\theta)$$
$$= \langle \mathbf{w}_{PC}(\phi), \mathbf{w}_{AC}(\phi) \rangle \otimes \langle \mathbf{w}_{PC}(\theta), \mathbf{w}_{AC}(\theta) \rangle, \tag{3}$$

where, $\mathbf{w}_{PC}(\phi)$ and $\mathbf{w}_{PC}(\theta)$ represent the column vectors of the main codebook in the azimuth and elevation angle domains, respectively. $\mathbf{w}_{AC}(\phi)$ and $\mathbf{w}_{AC}(\theta)$ represent the column vectors in the

azimuth and elevation domains in the auxiliary codebook, respectively. $A \otimes B$ means the Kronecker product of matrices A and B. The primary codebook and auxiliary codebook are designed to be the Kroneck product of the $\phi$ domain and $\theta$ domain codebooks.

### 2.2. System Model

The signal received at the receiver can be written as:

$$\mathbf{y} = \sqrt{P_{tot}} \mathbf{w}_R^H \mathbf{H} \mathbf{w}_T \mathbf{s} + \mathbf{w}_R^H \mathbf{n}, \tag{4}$$

where $\mathbf{s}$ denotes the transmission symbol with the unit power, $\mathbf{w}_T$ and $\mathbf{w}_R$ are the $N_T \times 1$ transmit and $N_R \times 1$ receive beamforming vectors, respectively. $\mathbf{n}$ is the Gaussian noise vector of power $N_0$, ie., $E\left(\mathbf{n}\mathbf{n}^H\right) = N_0 \mathbf{I}_{N_R}$ and $\mathbf{H}$ is the $N_R \times N_T$ channel matrix. The channel model can be modeled by a combination of a line of sight (LOS) path and several non-line of sight (NLOS) paths, as in [20]:

$$\begin{aligned}
\mathbf{H} &= \sqrt{\frac{\kappa}{\kappa+1}} \mathbf{a}_R \left(\theta_R^0, \phi_R^0\right) \mathbf{a}_T^H \left(\theta_T^0, \phi_T^0\right) \\
&+ \sqrt{\frac{1}{I(\kappa+1)}} \sum_{i=1}^{I} \alpha_i \mathbf{a}_R \left(\theta_R^i, \phi_R^i\right) \mathbf{a}_T^H \left(\theta_T^i, \phi_T^i\right),
\end{aligned} \tag{5}$$

where $\kappa$ indicates the Ricean factor, $\alpha_i \sim CN(0,1)$ is the complex channel gain and $I$ is the number of NLOS paths. Each ray is represented by a pair of two angles $(\phi, \theta)$, as shown in Figure 1. $\mathbf{a}_{\{R/T\}}(\cdot, \cdot)$ is a $N_{\{R/T\}} \times 1$ vector representing the phase offset of the antenna elements introduced during reception/transmission. The average power transmitted from each transmit antenna is equal to $P_{tot}/N_T$, so the total radiated power transmitted from the transmitter is $P_{tot}$. After combining the signals of all receiving antenna elements, the effective received signal-to-noise ratio (SNR) is defined as:

$$\delta_R = P_{tot} \left| \mathbf{w}_R^H \mathbf{H} \mathbf{w}_T \right|^2 / N_0. \tag{6}$$

The total beamforming gain at the receiver can be expressed as [21]:

$$G_B = \left| \mathbf{w}_R^H \mathbf{H} \mathbf{w}_T \right|^2. \tag{7}$$

## 3. The Two-Stage Codebook Design

### 3.1. Primary Codebook Design

Although most of the literature proposes different codebook designs based on the ULA model, little is known about the work of 2D antenna arrays (especially URA). This paper combines the idea of beam steering and layered codebook design to form a layered codebook in three-dimensional space as the primary search codebook. Based on the azimuth range $[0, 2\pi]$ and the elevation angle $\left[-\frac{\pi}{2}, \frac{\pi}{2}\right]$ in the actual millimeter wave communication, respectively, we design the azimuth and elevation angle codebooks separately, and then do the Kronecker product to get the final complete codebook. Taking the azimuth angle as an example, we can see Table 2, we first design the codebook in this dimension and organize it into a binary tree. The codebook design should meet the following three principles [9]:

- The primary codebook should be hierarchical.
- The corresponding beams of all AWVs (array weighted vectors) on each layer should cover the entire space together.

- The half power beam width (HPBW) of a certain weight vector of this layer covers the HPBW of the corresponding two adjacent weight vectors of the next layer, namely:

$$
\begin{aligned}
HPBW\left\{\mathbf{w}_\phi\left(k,i\right)\right\} &= HPBW\left\{\mathbf{w}_\phi\left(k+1,2i\right)\right\} \\
&\cup HPBW\left\{\mathbf{w}_\phi\left(k+1,2i-1\right)\right\},
\end{aligned}
\tag{8}
$$

where $\mathbf{w}_\phi\left(k,i\right)$ is the $i-th$ weight vector of $k-th$ layer and $HPBW\left\{\mathbf{w}_\phi\left(k,i\right)\right\}$ represents the half-power beamwidth of $\mathbf{w}_\phi\left(k,i\right)$. Based on these three principles, the structure of the binary codebook is listed in Table 2.

**Table 2.** Layered primary codebook design.

| Layers | AWVs of Each Layer in Primary Codebook |
|:---:|:---:|
| 1 | $\langle\mathbf{w}\left(1,1\right)\rangle$ |
| 2 | $\langle\mathbf{w}\left(2,1\right),\mathbf{w}\left(2,2\right)\rangle$ |
| 3 | $\langle\mathbf{w}\left(3,1\right),\mathbf{w}\left(3,2\right),\mathbf{w}\left(3,3\right),\mathbf{w}\left(3,4\right)\rangle$ |
| $\vdots$ | $\vdots$ |
| $k$ | $\left\langle\mathbf{w}\left(k,1\right),\mathbf{w}\left(k,2\right),...,\mathbf{w}\left(k,2^{k-1}\right)\right\rangle$ |
| $\vdots$ | $\vdots$ |
| $\log_2 N+1$ | $\langle\mathbf{w}\left(\log_2 N+1,1\right),...,\mathbf{w}\left(\log_2 N+1,N\right)\rangle$ |

The antenna has $N$ row elements in $x$ axis direction, and the multi-resolution codebook is designed to have $\lceil\log_2 N\rceil+1$ layers, with several AWVs per layer being predefined by their expected beam coverage. The $k-th$ layer codebook $\left\langle\mathbf{w}_\phi\left(k,1\right),\mathbf{w}_\phi\left(k,2\right),\mathbf{w}_\phi\left(k,3\right),...,\mathbf{w}_\phi\left(k,2^{k-1}\right)\right\rangle$ evenly covers the entire space $\phi=[0,2\pi]$. The $k-th$ layer will have $2^{k-1}$ weight vectors, we can evenly sample the space $[0,2\pi]$ at intervals $\frac{2\pi}{2^{k-1}}$. The $i-th$ beam of the $k-th$ layer in the codebook is sampled at: $\phi\left(k,i\right)=\frac{\pi(2i-1)}{2^{k-1}}$. The antenna array response vector is $\mathbf{a}\left(N,\phi\right)=\frac{1}{\sqrt{N}}e^{-j\frac{2\pi}{\lambda}d_x m_x \sin\phi\sin\theta}$, and the weight vector can be obtained by using the steering vector function. The weight vector of the receiver is finally set to:

$$
\mathbf{w}_\phi\left(k,i\right)=\left[\mathbf{a}_\phi^T\left(\phi\left(k,i\right),2^{k-1}\right),\left(\mathbf{0}_{N-2^{k-1}}\right)\right]^T.
\tag{9}
$$

Similarly, in the dimension of the elevation angle, the codebook of $k-th$ layer uniformly is $\left\langle\mathbf{w}_\theta\left(k,1\right),\mathbf{w}_\theta\left(k,2\right),\mathbf{w}_\theta\left(k,3\right),...,\mathbf{w}_\theta\left(k,2^{k-1}\right)\right\rangle$ and it covers the elevation space $\theta=\left[-\frac{\pi}{2},\frac{\pi}{2}\right]$. We sample at even intervals within the range and have $\theta\left(k,i\right)=-\frac{\pi}{2^k}+(2i-1)\frac{\pi}{2^k}$, so the antenna array response vector is $\mathbf{a}\left(M,\theta\right)=\frac{1}{\sqrt{M}}e^{-j\frac{2\pi}{\lambda}d_y m_y \cos\theta}$, $m_y=0,1,2,...,M-1$, and the final vector of the elevation angle dimension is set to:

$$
\mathbf{w}_\theta\left(k,i\right)=\left[\mathbf{w}_\theta^T\left(\theta\left(k,i\right),2^{k-1}\right),\left(\mathbf{0}_{M-2^{k-1}}\right)\right]^T.
\tag{10}
$$

Here, $\mathbf{a}\left(\cdot\right)$ is the array response vector. The specific expression is shown in Equation (2). The AWV at the transmitter end can be set similarly. Take the receiver side antenna setting as an example, $N=M=16$, $d_x=d_y=\frac{\lambda}{2}$. We use beam steering to generate a beam $\left(\phi,\theta\right)$ which is centered at angle $\left(-20,-20\right)$ and the 3D beam pattern is shown in Figure 2.

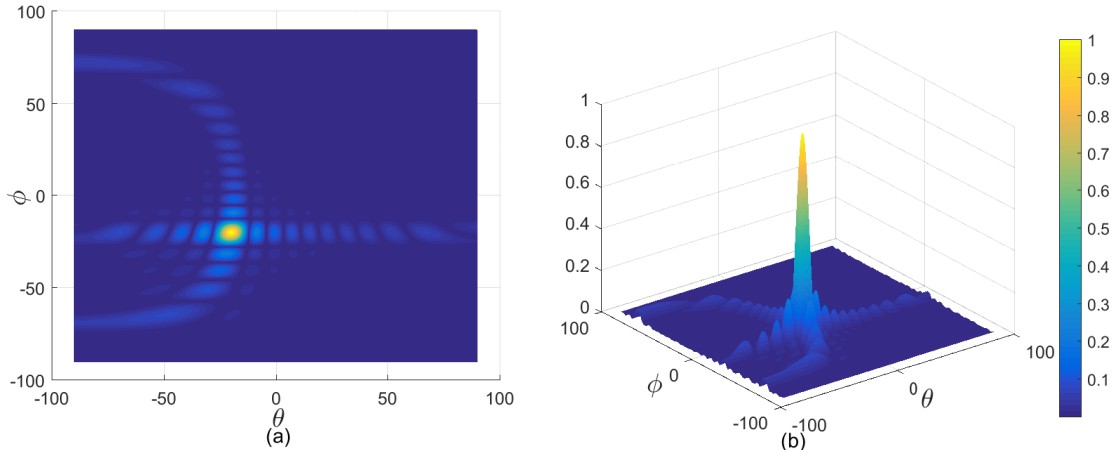

**Figure 2.** 3D beam pattern diagram, $N = M = 16$ (**a**) Top view (**b**) 3D perspective view.

For the designed primary codebook, the above three principles are met in the two dimensions of azimuth and elevation, respectively. Then a layered codebook based on the binary tree is formed in both dimensions, and finally the hierarchical beams are formed in the entire 3D space. We do not change the beam coverage width of the direction $\theta$, and change the beam coverage in the $\phi$ angle dimension by changing the parameters in $x$ axis direction to activate and deactivate the different numbers of antennas. Figure 3 is a schematic diagram of 3D beam coverage. The coverage of $\theta$ angle is unchanged, and in $\phi$ angular dimension, the 3rd and 4th beams $\mathbf{w}\,(4,3)\,,\mathbf{w}\,(4,4)$ of the fourth layer codebook in  3b cover the 5,6,7,8th beam $\mathbf{w}\,(5,5)\,,\mathbf{w}\,(5,6)\,,\mathbf{w}\,(5,7)\,,\mathbf{w}\,(5,8)$ in fifth layer codebook in Figure 3a.

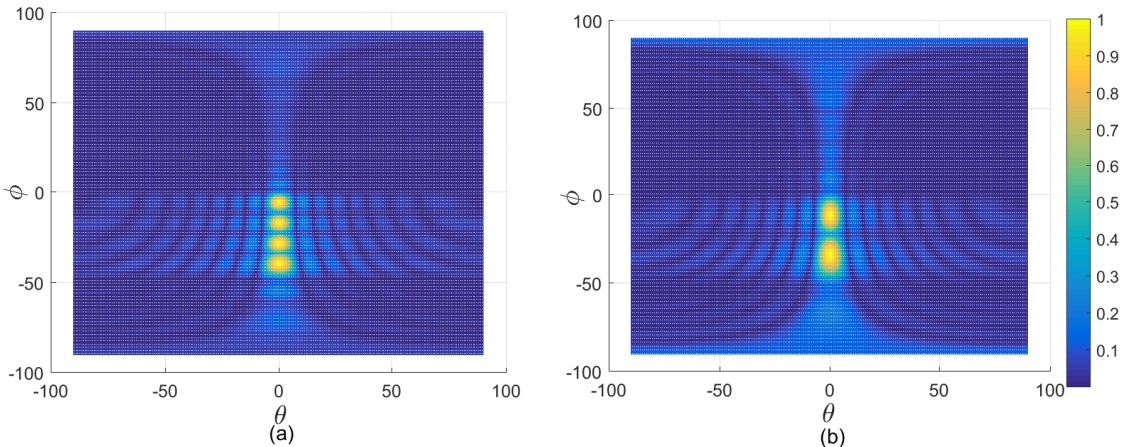

**Figure 3.**　　3D beam coverage diagram for $\phi$ dimension (**a**) $N = M = 16$, $\mathbf{w}\,(5,5)\,,\mathbf{w}\,(5,6)\,,\mathbf{w}\,(5,7)\,,\mathbf{w}\,(5,8)$; (**b**) $M = 16, N = 8, \mathbf{w}\,(4,3)\,,\mathbf{w}\,(4,4)$.

### 3.2. Auxiliary Codebook Design

In the primary codebook, there are $N$ row elements in $x$ direction and $M$ row elements in $y$ direction, thus forming an $N \times M$ antenna array. The maximum angular accuracy of the designed codebook is at the last level. At this time, the entire spatial space has $N$ beams in azimuth dimension and $M$ beams in elevtion dimension. The angular precision in these two dimensions can be $\frac{2\pi}{N}$ and $\frac{\pi}{M}$. This means that the deviations between the true angle and the codebook setting are in the range $\frac{2\pi}{N}$ and $\frac{\pi}{M}$ in azimuth and elevation dimension, respectively. To further improve the receive SNR and angular resolution, we increase the resolution to $\frac{1}{\alpha N_{beam}}, \alpha > 1, N_{beam} = N, M$. Here $\alpha$ is the refinement

factor, $N_{beam}$ is the number of the refined beams that can be achieved in the last layer of codebook. At the end of the UE, the weight vector in the codebook can be set to [12]:

$$
\begin{aligned}
\mathbf{w}_{\phi}^{AC}(i) &= \mathbf{a}_{\phi}(N, i), i = 1, 2, ..., \alpha N, \\
\mathbf{w}_{\theta}^{AC}(i) &= \mathbf{a}_{\theta}(M, i), i = 1, 2, ..., \alpha M.
\end{aligned}
\tag{11}
$$

The entire space originally covered by $N_{beam}$ beams is now covered by $\alpha N_{beam}$ beams. In the actual beam search process, we first use primary codebook to generate the basic directional beam. By expanding the size of the primary codebook, the auxiliary codebook can produce beams centered at the primary beam with a higher resolution. Figure 4 shows the beam pattern of the primary codebook and auxiliary codebook. Taking the azimuth dimension as an example, we let $M = 9, \alpha = 4$, and first create a main direction beam, which is indicated by the red line in Figure 4. The thinner beams produced by the auxiliary codebook are centered on the main beam direction.

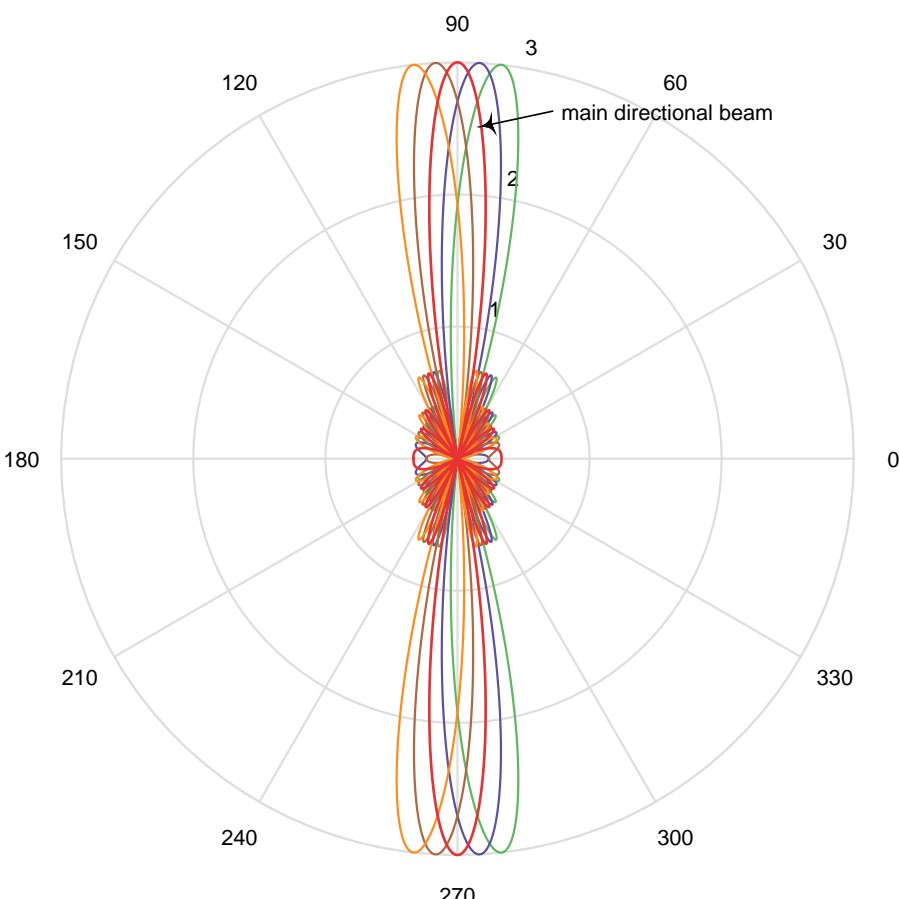

**Figure 4.** Last layer of beam pattern in primary codebook and auxiliary codebook when $N = 9, \alpha = 4$.

## 4. Fast Search Algorithm

For the beam training algorithms based on the hierarchical codebook, we will further reduce the complexity of the beam search by reducing the number of weight vectors that each layer needs to calculate. In this paper, a BSL-VT (binary search like method based on variable threshold) 3D beam search scheme is proposed based on the binary tree structure codebook to reduce the search time. By setting the upper and lower variable thresholds, the scheme can distinguish between the optimal beam vector and the non-optimal beam vector in most cases [9], thereby reducing the number of beam training in each layer of the codebook. The two-stage 3D beam access scheme proposed herein includes a primary beam search and an auxiliary beam search. In order to reduce the training overhead, we use

the multi-resolution binary tree search strategy in primary beam search stage to complete the azimuth and elevation angle dimension search in turn; in the second level training stage, the refined codebook is used, and the beam search is more precise. Auxiliary search also includes azimuth and elevation angle search. For the sake of simplicity, only the training process in the azimuth is described in the following sections. The beam search process in the elevation angle dimension can be analogized.

*4.1. 3D BSL-VT Primary Search*

A binary tree structure codebook is used, and a threshold mechanism is used to reduce the number of searches. The core of the 3D BSL-VT algorithm is the SNR upper and lower factor settings. Although the SNR of each layer codebook implementation is different, there is a rule:

- Rule 1: In a binary tree search, the number of active antennas in codebook at the transmitter (receiver) is twice that of the upper layer. The best AWV in this layer can achieve a SNR which is approximately twice the SNR obtained in the upper layer. With $\gamma_k$ means the best SNR achieved in the $k - th$ layer, we have:

$$\gamma_k > \gamma_{k-1}, \gamma_k \approx 2\gamma_{k-1}. \tag{12}$$

To qualitatively describe this rule, we set a variable threshold. The initial threshold factor can be set to the empirical value $\eta_1 = 1.6, \eta_2 = 0.8$. Algorithm 1 is an analysis of 3D BSL-VT primary search in $\phi$ dimension. The elevation angle dimension can be analyzed by example.

---

**Algorithm 1:** 3D BSL-VT primary search for $\phi$.

---

(1) Initialization Starting from the user equipment (UE) side, we first use the codeword $\mathbf{w}_\phi^{PC}(1,1), \mathbf{w}_\theta^{PC}(1,1)$, on the elevation and azimuth angle dimensions. Therefore, we use the codeword $\mathbf{w} = \mathbf{w}_\phi^{PC}(1,1) \otimes \mathbf{w}_\theta^{PC}(1,1)$ to receive a fixed training signal from the base station(BS) and calculate the received SNR $\gamma_0$. And set the following parameters: $k = 1, i = 1, \gamma_{ref} = \gamma_0, \eta_1 = 1.6, \eta_2 = 0.8$.

(2) Step 1 search

The BS always sends the training signal at a fixed AWV. We use the codeword $\mathbf{w} = \mathbf{w}_\phi^{PC}(2,1) \otimes \mathbf{w}_\theta^{PC}(1,1)$ to receive the signal and then calculate the corresponding SNR $\gamma_1$. If $\gamma_1 \geq \gamma_{ref} \cdot \eta_1$, then $\gamma_{ref} = \gamma_1, i = 2i$, otherwise there is $\gamma_{ref} = \gamma_{ref} \cdot \eta_1, i = 2i - 1$.

(3) Step $k$ search

1. UE receives the training signal using the following AWV: $\mathbf{w}_r = \mathbf{w}_\phi^{PC}(k+1,2i) \otimes \mathbf{w}_\theta^{PC}(1,1)$. To calculate the received SNR, we only calculate the SNR corresponding to the codeword $\mathbf{w}_r$, denoted as $\gamma_k$. If $\gamma_k \geq \eta_1 \cdot \gamma_{ref}$, then $i = 2i$, and update the threshold: $\gamma_{ref} = \gamma_k$; otherwise if $\gamma_k \leq \eta_2 \cdot \gamma_{ref}$; then there is $i = 2i - 1, \gamma_{ref} = \eta_1 \cdot \gamma_{ref}$.
2. If SNR is not in the judgment interval, then we detect once more, that is, we calculate the SNR related of the codeword $\mathbf{w}_r = \mathbf{w}_\phi^{PC}(k+1,2i+1) \otimes \mathbf{w}_\theta^{PC}(1,1)$ and denote $\tilde{\gamma}_k$. If there is $\tilde{\gamma}_k > \gamma_k$, then $i = 2i - 1, \gamma_{ref=\tilde{\gamma}_k}$; If there is $\tilde{\gamma}_k \leq \gamma_k$, then $i = 2i, \gamma_{ref=\tilde{\gamma}_k}$;

(4)Result We let $i_\phi^* = i$; the best AWV index of the codebook in azimuth dimension obtained in primary search process is $i_\phi^* = i$.

---

When scanning the elevation angle space, we use the best codebook index of $i_\phi^* = i$ that was found in azimuth dimension. In the initialization phase, we use codewords $\mathbf{w} = \mathbf{w}_\phi^{PC}\left(\log_2 N + 1, i_\phi^*\right) \otimes \mathbf{w}_\theta^{PC}(1,1)$ to receive signals. In the next search steps, our azimuth codebook will still use the best codeword found in the first phase. The beam training process in the elevation angle direction is similar

to the first phase. Finally we can find the best beamforming vector $\mathbf{w}_{opt} = \mathbf{w}_{\phi}^{PC}\left(\log_2 N + 1, i_{\phi}^*\right) \otimes \mathbf{w}_{\theta}^{PC}\left(\log_2 N + 1, i_{\theta}^*\right)$. Then the primary search process ends.

### 4.2. 3D BSL-VT Auxiliary Search

Auxiliary search is initiated after the main search to achieve a higher precision AWV pair. After the primary beam search, we get the best AWV: $\mathbf{w}_{opt} = \mathbf{w}_{\phi}^{PC}\left(\log_2 N + 1, i_{\phi}^*\right) \otimes \mathbf{w}_{\theta}^{PC}\left(\log_2 N + 1, i_{\theta}^*\right)$. The entire spatial azimuth and elevation angle dimensions are divided into $N$ and $M$ beams. To further increase the angular resolution, we will increase the resolution to $\frac{1}{\alpha N_{beam}}, \alpha > 1, N_{beam} = N, M$. In both dimensions, the initial AWV pair $\left(\mathbf{w}_{\phi}^*, \mathbf{w}_{\theta}^*\right)$ and its index is $\left(i_{\phi}^*, i_{\theta}^*\right)$. According to (11), the initial AWV in the auxiliary codebook is set to $\tilde{\mathbf{w}}_{ini} = \mathbf{w}_{\phi}^{AC}\left(\log_2 N + 1, \alpha i_{\phi}^*\right) \otimes \mathbf{w}_{\theta}^{AC}\left(\log_2 M + 1, \alpha i_{\theta}^*\right)$. Similarly, auxiliary search has two phases. In the first phase, we scan the azimuth space to find the best codeword $\tilde{\mathbf{w}}_{\phi}^*$ and its index $m_{\phi}^*$. In the second phase, the elevation angle space is scanned to find the best codeword $\tilde{\mathbf{w}}_{\theta}^*$ and index $m_{\theta}^*$. Here, we take the UE side for example, and the auxiliary search beam vector index ranges are $m_{\phi}^* \in \left[\alpha i_{\phi}^* - \frac{\alpha}{2}, \alpha i_{\phi}^* + \frac{\alpha}{2}\right]$ and $m_{\theta}^* \in \left[\alpha i_{\theta}^* - \frac{\alpha}{2}, \alpha i_{\theta}^* + \frac{\alpha}{2}\right]$. Compared to the main search, we can create a finer beam Similar to [9], this article use the binary search scheme to find the best beam. The scan of the azimuth angle in the auxiliary search is described in Algorithm 2.

---

**Algorithm 2:** 3D BSL-VT auxiliary search for $\phi$.

---

(1) Initialization

The UE receives the fixed training signal from the BS by using the codeword $\tilde{w} = w_{\phi}^{AC}\left(\alpha i_{\phi}^*\right) \otimes w_{\theta}^{AC}\left(\alpha i_{\theta}^*\right)$, and the index range of the codeword of the UE-side during refinement search is set to $m_L = \alpha i_{\phi}^* - \frac{\alpha}{2}, m_R = \alpha i_{\phi}^* + \frac{\alpha}{2}$.

(2) Search process

Perform the following search process $\log_2 \alpha$ times and then stop:

1. If $m_L = i_{\phi}^* \alpha$ or $m_R = i_{\phi}^* \alpha$, we can take advantage of the best SNR obtained in the primary search, we let $\gamma_L = \gamma^*$ or $\gamma_R = \gamma^*$ directly.
2. The receiver receives the signals using the AWVs corresponding to the $m_L - th$ and $m_R - th$ index in the auxiliary codebook, and calculates the received SNR $\gamma_L$ and $\gamma_R$ respectively.
3. If $\gamma_L \geq \gamma_R$, then let $m_R = \left[\frac{m_R + m_L}{2}\right]$, $[\cdot]$ means round down the calculation; otherwise $m_L = \left[\frac{m_R + m_L}{2}\right]$;

(3) Result

Returns the refined best codeword $\tilde{w}_{\phi}^*$ and its index $m_{\phi}^*$ obtained during the auxiliary search with the auxiliary codebook.

---

## 5. Experimental Verification

In this section, we will provide simulation results to demonstrate the effectiveness of our proposed two-stage codebook design and beam training scheme. Referring to Reference [11], millimeter-wave channel modeling is described in Section 2. Detailed simulation parameters are provided in Table 3.

**Table 3.** Simulation parameters setting.

| Parameter | Value |
|---|---|
| Central frequency | 28 GHz |
| Element space | $0.5\lambda$ |
| Channel model | Channel model in [11] |
| Antenna array | URA$(N = M)$ |
| BS antenna number | $(N = M)$ from 8 to 256 |
| UE antenna number | $(N = M)$ from 8 to 256 |
| AoA/AoD | uniformly distributed in $[0, 2\pi]$ |
| ZoA/ZoD | uniformly distributed in $\left[-\frac{\pi}{2}, \frac{\pi}{2}\right]$ |
| Number of users | 1 |
| Data stream per user | 1 |
| Refinement factor | $\alpha = 4$ |

### 5.1. The Necessity of The Two-Stage Codebook

In order to verify the validity of the auxiliary codebook and the auxiliary search, we compared the beamforming gain obtained by the 3D BSL-VT algorithm using the auxiliary search and using only the primary search. We assume that the BS side transmits fixed training beam periodically. At the UE receiving end, we configure the URA array, where $N = M = 32$. The refinement factor in the auxiliary codebook is set to $\alpha = 4$. There will be one LOS path in the communication channel with approximately 3 to 5 NLOS paths. The average power setting of the NLOS path is 15 dB lower than the average power of the LOS path, which is a very typical case in millimeter wave communication. We start the primary search, firstly search from the azimuth angle dimension, and then scan the elevation angle dimension to obtain the best beam index and the final cumulative beamforming gain $B_{coarse}$. Then, the auxiliary search is started, and the beam is further refined by expanding the codebook size in two dimensions to obtain the final refined beamforming gain $B_{finer}$.

Figure 5 shows how the beam gain of the receiver varies with the search steps. During the primary beam search process, the first stage of 3D BSL-VT is the azimuth search refinement, after $\log_2 N = \log_2 32 = 5$ steps the best beam and its index of the azimuthal dimension are found. Then we refines the beam in the elevation angle dimension using $\log_2 M = \log_2 32 = 5$ steps and finds the best beam vector. A total of 10 search steps are required for the entire primary search. Auxiliary search is also divided into two phases, which requires a total number of $2\log_2 \alpha = 4$ searches, and we merge the auxiliary search into index 11 on the graph. Index 11 in Figure 5 is the result of the auxiliary search. It can be seen that the gain value is slightly better than the gain contained in the main search during index 1 to 10. In addition, as the search step increases, the beam gain of the array becomes higher and higher by continuously refining the beam resolution. This is consistent with the simulation results in the figure. In contrast, the red dashed line represents the maximum beamforming gain that can theoretically be obtained under the same channel simulation conditions. The Figure 5 also shows that the final gain is eventually very close to the upper limit. Therefore, the 3D BSL-VT algorithm performs well in finding the optimal weight vector.

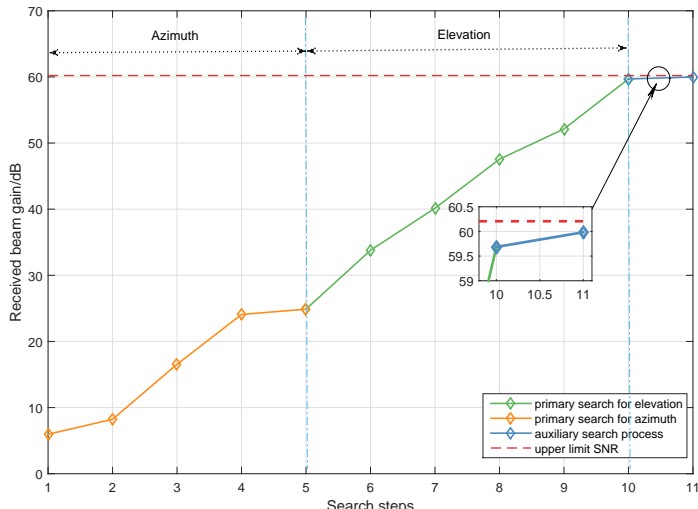

**Figure 5.** Beamforming gain during 3D BSL-VT beam training process when $N = M = 32, \alpha = 4$.

Figure 6 shows the simulation result of 100 times under the same condition. From Figure 6 we can see the beamforming gains using auxiliary search and only the primary search in 3D BSL-VT algorithm. As can be seen from this figure, in the simulation of up to 100 times, the result of the refinement search is always better than the rough search, and a higher cumulative gain can be obtained.

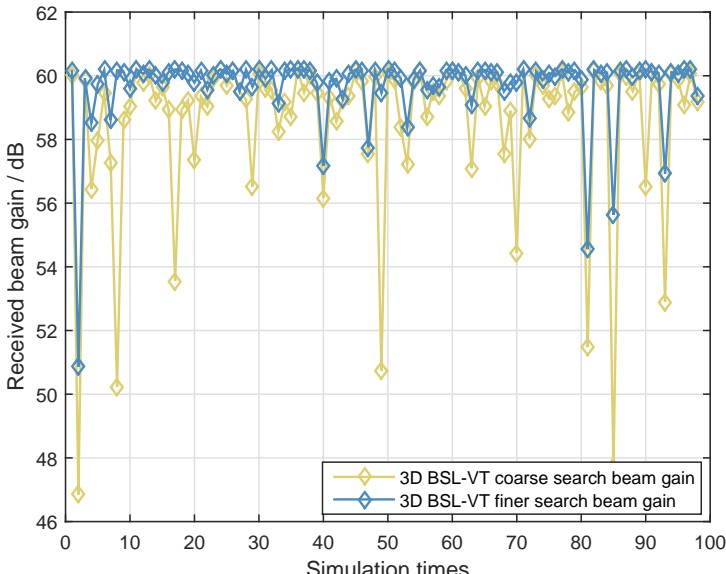

**Figure 6.** Maximum beamforming gain obtained by primary search and auxiliary search.

In fact, the purpose of designing different beam search algorithms is to reduce the beam training time as much as possible on the basis of ensuring the robustness of the beam search. It is assumed that in most cases the beam training process may find the best or relatively suboptimal communication beam pair, the beamforming gain under the same channel conditions is mainly related to the codebook design. Here, in order to verify the superiority of the beam gain of our two-stage codebook design, we compared with the remaining three typical codebook designs:

1.  IEEE 802.15.3c Codebook

    The IEEE 802.15.3c standard [4] uses only 4 phase shifts $(0°, 90°, 180°, 270°)$ to form a codebook $W$:

    $$w(m,k) = j^{floor\left\{\frac{m \times mod((k+K/2),K)}{K/4}\right\}}, m = 0, 1, ...M-1; k = 0, 1, ...K-1, \tag{13}$$

    where $k$ is the beam pattern index and $m$ is the element index. $M$ is the number of elements and $K$ is the total beam number.

2.  Beam-steering Codebook

    Beam-steering codebooks [6] have the same form as antenna response vectors, and can be parameterized by a simple angle. The angle is generated by quantizing the RF angle with several bits in the case of limited feedback. The $(m,k) - th$ antenna element can be defined as:

    $$w(m,k) = \exp\left\{j\pi m \sin\left\{\frac{2\pi k}{2^{Bits}}\right\}\right\}, m = 0, 1, ...M-1; k = 0, 1, ...2^{Bits}, \tag{14}$$

    where $Bits$ is the number of quantized bits.

3.  DFT Codebook

    The DFT codebook is another form of codebook defined by RF angle quantization [5]. More specifically, both the number of beam patterns and the number of phase shifts equal the number of elements $M$. The expression of the DFT codebook is given by:

    $$w(m,k) = \exp\left\{j\frac{2\pi mk}{M}\right\}, m = 0, 1, ...M-1; k = 0, 1, ...M-1. \tag{15}$$

    In the simulation experiment, we configure the $4 \times 4$ URA array totally 16 elements at the sending end. On the receiving end, the configured URA array has $M = N \in [2, 64]$. We explored the change in beam gain of various codebooks with different numbers of antenna arrays, and the results are shown in Figure 7. The beam gain results at each point in the figure are the average values obtained through 200 simulations under the same channel conditions.

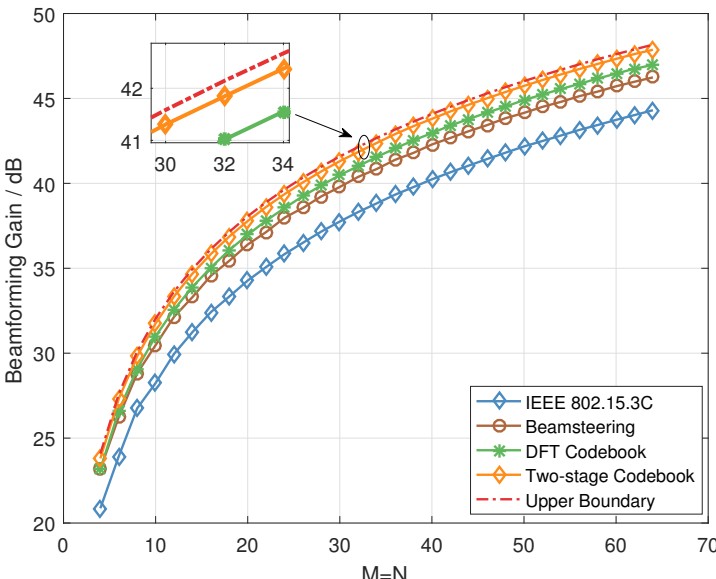

**Figure 7.** The Beamforming gains of the various codebooks.

Figure 7 shows that our proposed two-stage codebook outperforms other codebooks by about 2–3 dB. In fact the additional beamforming gain is obtained from the enhanced beam resolution due to the use of the auxiliary codebook in our proposed design. In particular, the beam gain of the codebook we mentioned is very close to the upper limit of the gain under the same channel conditions. The two-stage codebook design in this paper has superiority in obtaining better beforming gain.

## 5.2. Search Accuracy Verification

In order to verify that our beam search algorithm based on codebook design has better search accuracy, we performed 100 simulations under the same condition. Although exhaustive search strategy has high search complexity, it can always find the best communication beam pair. Figure 8 compares the cumulative gain obtained by the fine search in BSL-VT scheme with the beam gain achieved by exhaustive search algorithm(indicated by the red line). From the simulation results, the result value of almost every gain reaches the exhaustive search result which can be seen the upper limit. After statistical analysis, the number of times can be 6 when the beam gain is lower than the upper limit of 2 dB. If we take the value of 2 dB below the upper limit as the threshold value, we can think that the beam search accuracy is about up to 94% and the codebook design and beam training algorithm proposed in this paper have better robustness.

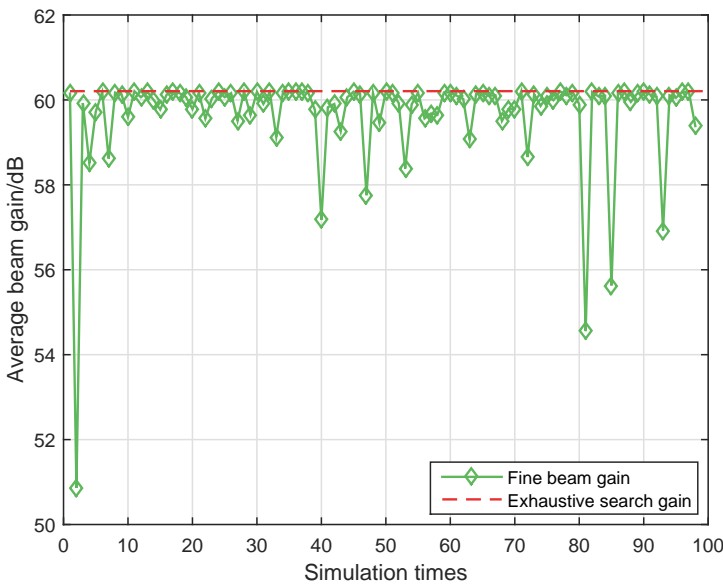

**Figure 8.** Cumulative gain obtained by the 3D BSL-VT algorithm, $N = M = 32, \alpha = 4$.

## 5.3. Comparison of Algorithm Complexity

Referring to Reference [21], we compared different initial access (IA) techniques from the perspective of signal access delay. Considering the signal duration $T_{sig} = 10\mu s$ and target overhead $\phi_{ov} = 5\%$, the time between two consecutive time slot transmissions must be at least $T_{per} = T_{sig}/\phi_{ov} = 200\mu s$. In the whole beam search process, we believe that the number of timeslots for general beam training is much larger than the feedback timeslots, and there are only several timeslots to feedback the beam index which needs only 1 to 2 bits. The beam switching time is at the nanosecond level, which can be ignored. So similar to other literatures, this paper focus on the number of beam training times, which is is almost equivalent to the beam access time. Since the IA process requires $N_s$ transmission slots for beam search, we calculate the search complexity as $N_s$ slots times. Now, we have configured URA on both sides of the transceiver, and $N = M$, the array has the same number of rows and columns. When searching for the azimuth and elevation angle, the codebook has $\log_2 N$ and $\log_2 M$ layers, respectively. The threshold-based method only needs to detect once for each layer. Both the sender and

the receiver must perform, so the number of primary search probes is $2 \times (\log_2 N + \log_2 M)$. Refine the search, the number of times at the transmitter or receiver is $2 \log_2 \alpha$ (two dimensions). The total detection time based on 3D BSL-VT algorithm can be expressed as:

$$T_{BSL-VT} = 2 \times (\log_2 N + \log_2 M) + 2 \times 2 \log_2 \alpha. \tag{16}$$

For the same analysis, we extend some algorithms to 3D beam search, assuming that the finer search in these algorithms uses exhaustive search, the total number of search for the BSL algorithm [9] is:

$$T_{BSL} = 2 \times (2 \log_2 N + 2 \log_2 M) + 2 \times 2 \log_2 \alpha. \tag{17}$$

Reference [16] uses a new beam access method based on the mainlobe overlapping state of a training beam and the complexity can also be reduced to a logarithmic level:

$$T_{[16]} = 2 \times (\log_2 N + \log_2 M) + 2 \times \alpha^2. \tag{18}$$

Compared with the BSL algorithm, the complexity of the 3D beam search algorithm in this paper is reduced by nearly half. In order to compare with other search strategies, we also derived the search complexity of related algorithms. Suppose the transmitter and receiver take exactly the same antenna configuration, and we let $N = M$. We only consider the azimuth angle dimension, $N_{sec\,tor}$ is the number of sectors. Here we let $N_{sec\,tor} = 4$, $N_{beam}$ is the number of beams in each sector, and there are $N = N_{sec\,tor} \times N_{beam}$ communication beams in the azimuth direction. According to the literature [4], the number of searches at one dimension is $N_{sec\,tor}^2 + N^2 / N_{sec\,tor}^2$. Combining the azimuth and elevation angles, the auxiliary search uses an exhaustive strategy. The search complexity of IEEE 802.15.3c is:

$$T_{3c} = 2 \times \left( 2N_{sec\,tor}^2 + \frac{N^2 + M^2}{N_{sec\,tor}^2} \right) + 2 \times 2\alpha^2. \tag{19}$$

The search complexity is $O\left(N^2\right)$. Similar to the 3c standard above, the IEEE 802.11ad method has:

$$T_{ad} = 2 \times \left( 2N_{sec\,tor}^2 + N + M \right) + 2 \times 2\alpha^2. \tag{20}$$

Exhaustive search is the simplest search algorithm [13]. The search complexity is:

$$T_{exhaustive} = 2 \times \left( N^2 + M^2 + 2\alpha^2 \right). \tag{21}$$

Reference [12] proposed a two-stage beam search algorithm for 3D MIMO millimeter-wave communication scenarios. The entire beam search process includes a coarse search and a finer search. The coarse search process adopts hierarchical codebook design and binary search algorithm so the search steps is $2 \times (2 \log_2 N + 2 \log_2 M)$. The finer search uses a strategy of approximate exhaustive search algorithm and $2 \times \left\lceil (\alpha - 1)^2 / 4 + 9 \right\rceil$ steps are needed [12]. The search complexity of the two-stage scheme in Reference [12] can be:

$$T_{[12]} = 2 \times (2 \log_2 N + 2 \log_2 M) + 2 \times \left\lceil (\alpha - 1)^2 / 4 + 9 \right\rceil \tag{22}$$

Compared with the training mechanism proposed in Reference [12], our two-stage beam search algorithm is improved in the primary search and auxiliary search stages, respectively. Based on the change law of the received SNR, The primary search can reduce the binary search method in Reference [12] to a single detection in each layer of the codebook. The auxiliary search changes the approximate exhaustive search in Reference [12] to a binary detection. Both theoretical analysis and experimental results verify that compared with the scheme in Reference [12], the algorithm in this paper can reduce the search complexity by approximately half.

In Figure 9, we plot the search complexity curves for all of the above beam search schemes. For the URA antenna, we have $N = M$, the abscissa in the graph indicates the number of arrays. We have $N = M = 32$ for example, the thinnest beam angle width we can achieve in the azimuth and elevation dimensions is about $\frac{2\pi}{32} = 11.25°$ and $\frac{\pi}{32} = 5.63°$, respectively. The search times for exhaustive strategy, 802.15.3c, 802.11ad are 4224, 384, 258, respectively. BSL scheme and method in Reference [16] can reduce the steps to 48 and 52, especially the two-stage scheme in Reference [12] needs 62 steps, but our 3D BSL-VT algorithm only needs 28 steps meaning $200\mu s \times 28 = 5.6ms$ beam access time. If we only use coarse search, we only need 20 steps. As we analyzed above compared with the two-stage method in Reference [12], our fast beam search algorithm based on the variable SNR threshold can reduce the search time by nearly half. It can be seen from Figure 9 that the exhaustive search algorithm has the highest time complexity and some traditional hierarchical schemes, such as IEEE 802.15.3c and IEEE 802.11ad, have lower complexity. What's more, the method in Reference [16] and the BSL scheme can further achieve better performance. On this basis, our proposed 3D BSL-VT algorithm can reduce the search time even more on the basis of achieving the same angular precision. It can be seen from the figure that, compared with the general beam training algorithm, the training time increases sharply with the increase of the number of antennas. The proposed strategy can maintain the training time at a small level in a large-scale MIMO scenario.

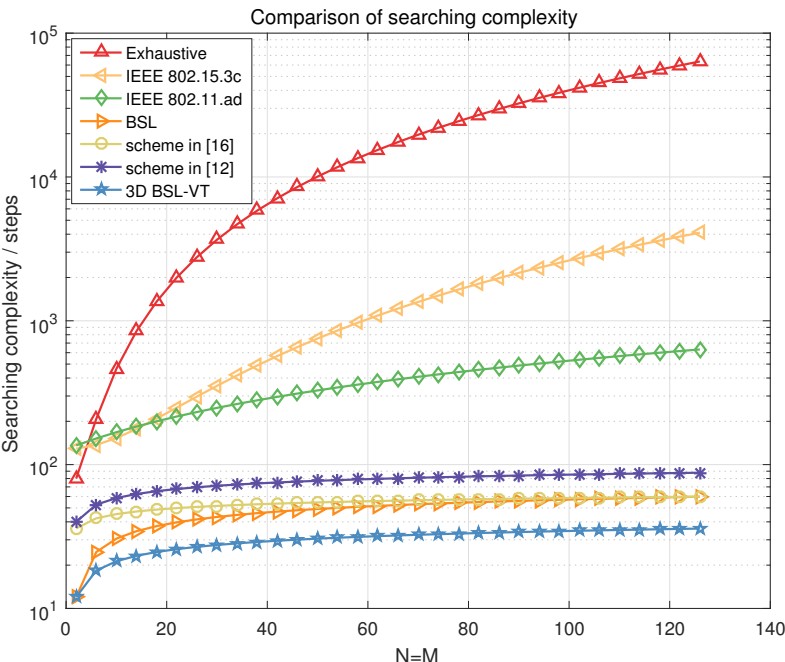

**Figure 9.** Comparison of the complexity between different beam training algorithms.

## 6. Conclusions

The focus of this paper is the design of a new two-stage codebook and associated beam search scheme for analog beamforming in a millimeter-wave 3D MIMO system. There are few papers in this field, and the 3D MIMO beamforming is complicated. The proposed codebook design can be well adapted to the URA model and the fast beam search algorithm can achieve a low search complexity in the 3D beam training process. Theoretical analysis and simulation results show that the proposed codebook design and beam search algorithm have good practicability and flexibility. Compared with the existing search algorithm, the search time can be greatly reduced and the search accuracy is good. This work has a certain significance for the development of the future 5G communication system. One drawback of the proposed scheme is that it may not work well in complicated NLoS scenarios. Therefore, our future work will study a more appropriate method for NLoS scenarios.

**Author Contributions:** Conceptualization, W.L.; Methodology, W.L.; Project administration, Z.P.; Software, W.L.; Supervision, Z.P.; Writing—original draft, W.L. All authors have read and agreed to the published version of the manuscript.

**Funding:** This research received no external funding.

**Acknowledgments:** Some of the ideas and opinions are from Liu Yang and Zhang in the laboratory. Thank you for their suggestions.

**Conflicts of Interest:** The authors declare no conflict of interest.

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
