# Peer review of "Two-Stage 3D Codebook Design and Fast Beam Search Algorithm for Millimeter-Wave Massive MIMO Systems"

_electronics, doi:10.3390/electronics9020302_

Round 1

Reviewer 1 Report

In this paper, a two-stage codebook and associated beam search scheme is proposed for analog beamforming in a millimeter-wave 3D MIMO system. The search complexity of the proposed scheme is compared with those of other existing training algorithms in terms of the number of steps. The reviewer asks authors to compare the complexities of training algorithms in terms of the number of feedbacks required to complete beam training algorithms. Normally, the process of reporting the information obtained at the MS to BS via feedback loop requires a large overhead. Also, the performance (beam gain) of the proposed scheme needs to be compared with those of other existing solutions.

The followings are typos:

The fifth line below Eq. (15) Line 268 in page 14

Reviewer 2 Report

The paper presents a two-stage 3D codebook design and beam training algorithm for mm-wave massive MIMO applications. The paper is well presented, structured. Some comments should be addressed by the author prior to further considerations: 

The “Author Contributions” section should be added. Please update the literature with the recent state of the arts. More references, especially from 2018 and 2019, should be included. It is better to add a table of Comparison between the proposed design/algorithm with recently reported ideas. Fort sizes of Figures 2 and 3 should be improved. Check also that all reference entry details are correct and listed according to a unified discipline. The title of the paper is very similar to “Two-Stage 3D Codebook Design and Beam Training for Millimeter-Wave Massive MIMO Systems", 2017 IEEE 85th Vehicular Technology Conference (VTC Spring). The difference and novelty should be discussed. Due to this point, the title also needs to be modified. Also, check again carefully about typo or misused figure/expression.

Author Response

Dear Editor and Reviewers:
Thank you for your letter and for the reviewers’ comments concerning our manuscript entitled “Two-stage 3D Codebook Design and Beam Training Algorithm for Millimeter-wave Massive MIMO Systems” (ID: electronics-703042). Those comments are all valuable and very helpful for revising and improving our paper, as well as the important guiding significance to our researches. We have studied comments carefully and have made correction which we hope meet with approval. Revised portion are marked in yellow in the paper. The main corrections in the paper and the responds to the reviewer’s comments are as flowing:

Responds to the reviewer’s comments:

Reviewer #2:

Response to comment: (The “Author Contributions” section should be added.)

Response: I add the “Author Contributions” section at the bottom of the paper at Line 316.

Response to comment: (Please update the literature with the recent state of the arts. More references, especially from 2018 and 2019, should be included. It is better to add a table of Comparison between the proposed design/algorithm with recently reported ideas.)

Response:

Combining with the literature in recent years, I appropriately added three new literatures [17-19]. I also added literature citations at the end of the article. A brief summary of the content of the new literature is found in Line 58 to 68. To briefly summarize the beam training algorithms mentioned above, referring to new add Table 1, we simply list the commonly used schemes at Table 1 and claim it at Line 68 to 70.

Response to comment: (Font sizes of Figures 2 and 3 should be improved.)

Response:

I have increased the font size of Figure 2 and 3 by 2 units, which can be seen clearly, but for the convenience of figure editing, I simply adjusted the position of the two sub-pictures in Figure 3, this has no effect on the content of the article.

Response to comment: (Check also that all reference entry details are correct and listed according to a unified discipline.)

Response:

I checked that the original reference 17 is a wrong reference and has been corrected. Rechecked citations for entire article after adding new citations.

Response to comment: (The title of the paper is very similar to “Two-Stage 3D Codebook Design and Beam Training for Millimeter-Wave Massive MIMO Systems", 2017 IEEE 85th Vehicular Technology Conference (VTC Spring). The difference and novelty should be discussed. Due to this point, the title also needs to be modified.)

Response:

In our intrduction, we have introduced the paper[12] at Line and 39 and 54: Two-Stage 3D Codebook Design and Beam Training for Millimeter-Wave Massive MIMO Systems", 2017 IEEE 85th Vehicular Technology Conference (VTC Spring). [12] proposed a two-stage beam search algorithm for 3D MIMO millimeter-wave communication scenarios, which can obtain a higher communication SNR at a lower training time through coarse search and finer search.

Compared with the training mechanism proposed in [12], our two-stage beam search algorithm is improved in the primary search and auxiliary search stages, respectively. Based on the change law of the received SNR, The primary search can reduce the binary search method in [12] to a single detection in each layer of the codebook. The auxiliary search changes the approximate exhaustive search in [12] to a binary detection. Both theoretical analysis and experimental results verify that compared with the schme in [12], the algorithm in this paper can reduce the search complexity by approximately half.

 Our beam search can cut the search time by neary half and have relative high search beam gain at the same time. Our method is actually fast, so to make diffeence, we Modify the title to: Two-stage 3D Codebook Design and Fast Beam Search Algorithm for Millimeter-wave Massive MIMO Systems. We have added some comparative analysis content in Line 281(below eq.21) to 287. In order to better compare the difference between [12] and our method, we analyzed the complexity of the beam training in [12] at line Line 281(below eq.21) to 287. and updated Figure 9, adding the curve of [12].

Response to comment: (Also, check again carefully about typo or misused figure/expression.)

Response: I carefully about typo or misused figure/expression and some changes can be seen at the change list below.

Reviewer #1:
1. Response to comment: (The reviewer asks authors to compare the complexities of training algorithms in terms of the number of feedbacks required to complete beam training algorithms. Normally, the process of reporting the information obtained at the MS to BS via feedback loop requires a large overhead.)
Response:

The general process of searching for a millimeter wave cell using a directional antenna is briefly described as follows:

Device 1 is the transmitting end. Device 2 is the receiving end and the pilot signal is received in several consecutive training slots. Device 2 can gradually refine the rceived beam and find the best beam index, only feedback the beam number to the transmitting end; Device 2 is the transmitting end using the best beam which has been found, and Device 1 is the receiving end to receive several consecutive pilot signals. Device 1 can find the best beam and feedback the number to Device 2.

3.The two devices can train the best beam pair for training.

Differrnt beam training scheme may have different feedback overhead. In general, during the whole beam search process, we believe that the number of timeslots for general beam training is much larger than the feedback timeslots, and there are only several timeslots to feedback the beam index which needs only 1 to 2 bits. The beam switching time is at the nanosecond level, which can be ignored. So similar to other literatures, this paper focus on the number of beam training times, which is is almost equivalent to the beam access time.

   I also made a simple explanation and modification in the paper at Line 281, 3 lines below the title “Comparison of Algorithm Complexity”.

Response to comment: (Also, the performance (beam gain) of the proposed scheme needs to be compared with those of other existing solutions)
Response:

In fact, the purpose of designing different beam search algorithms is to reduce the beam training time as much as possible on the basis of ensuring the robustness of the beam search. It is assumed that in most cases the beam training process may find the best or relatively suboptimal communication beam pair, the beamforming gain under the same channel conditions is mainly related to the codebook design. Also for some beam training schemes, like Exhaustive search,IEEE 802.15.3c and so on, these beam search methods can use several codebooks like 802.15.3c codebook, DFT codebooks and so on. Because Exhaustive search can always find the best communication beam pair, we compared our two-stage beam search with Exhaustive search here to prove that our beam search algorithm has high search accuracy. The result is shown in Figure 8.

So Here, in order to verify the superiority of the beam gain of our two-stage codebook design, we compared the remaining three typical codebook designs. We simply introduced several common codebooks and made some analysis from Line 248 to 269. The simulation results is show in the new made Figure7.

Response to comment: (The followings are typos:The fifth line below Eq. (15) Line 268 in page 14)

Response:

The error is: the number ofbeams ,we change the typo to: the number of beams.

Other changes:
1. Line 281, 8 lines below the title “Comparison of Algorithm Complexity”

 ‘Time’ and ‘since’ , the two words needs a space between them. It has been corrected.
2. Line 89, spell error : “veriy” was corrected to “veriy”.
3. Line 137,144, Because we add a new table:Table 1. So for Table 2, we change the reference “Table 1” to “Table 2”.

Line 193, Because we add a new table:Table 1. So for Table 3, we change the reference “Table 2” to “Table 3”. Line 213, Because we add a new table:Table 1. So for Table 4, we change the reference “Table 3” to “Table 4”. Line 217, Because we add a new table:Table 1. So for Table 5, we change the reference “Table 4” to “Table 5”. Line 274, Because we add a new figure:Figure 7. So for Figure 8, we change the reference “Figure 7” to “Figure 8”. Line 288,297, Because we add a new figure:Figure 7. So for Figure 9, we change the reference “Figure 8” to “Figure 9”.

We tried our best to improve the manuscript and made some changes in the manuscript. For the suggestions made by the reviewers, we have made corresponding changes in the paper and we have highlighted them in yellow in the above response.And here we list the other changes above and mark them in yellow in revised paper.
We appreciate for Editors/Reviewers’ warm work earnestly, and hope that the correction will meet with approval.Once again, thank you very much for your comments and suggestions.

Round 2

Reviewer 1 Report

The reference format does not follow the guidance in Instructions for Authors. Currently, it is written in IEEE format. 

Reviewer 2 Report

All comments and requests from first round of revision have been properly addressed and the manuscript is in my opinion acceptable for publication.